# Filter-based models of suppression in retinal ganglion cells: Comparison and generalization across species and stimuli

Neda Shahidi[1,2,3,4], Fernando Rozenblit[1,2], Mohammad H. Khani[1,2¤],
Helene M. Schreyer[1,2¤], Matthias Mietsch[5,6], Dario A. Protti[7], Tim Gollisch[1,2,8,9]*

**1** Department of Ophthalmology, University Medical Center Göttingen, Göttingen, Germany, **2** Bernstein Center for Computational Neuroscience Göttingen, Göttingen, Germany, **3** Georg-Elias-Müller-Institute for Psychology, Georg-August-Universität Göttingen, Göttingen, Germany, **4** Cognitive Neuroscience Lab, German Primate Center, Göttingen, Germany, **5** Laboratory Animal Science Unit, German Primate Center, Göttingen, Germany, **6** German Center for Cardiovascular Research, Partner Site Göttingen, Göttingen, Germany, **7** School of Medical Sciences (Neuroscience), The University of Sydney, Sydney, New South Wales, Australia, **8** Cluster of Excellence "Multiscale Bioimaging: from Molecular Machines to Networks of Excitable Cells" (MBExC), University of Göttingen, Göttingen, Germany, **9** Else Kröner Fresenius Center for Optogenetic Therapies, University Medical Center Göttingen, Göttingen, Germany

¤ Current address: Institute of Molecular and Clinical Ophthalmology Basel, Basel, Switzerland
* tim.gollisch@med.uni-goettingen.de

## Abstract

The dichotomy of excitation and suppression is one of the canonical mechanisms explaining the complexity of neural activity. Computational models of the interplay of excitation and suppression in single neurons aim at investigating how this interaction affects a neuron's spiking responses and shapes the encoding of sensory stimuli. Here, we compare the performance of three filter-based stimulus-encoding models for predicting retinal ganglion cell responses recorded from axolotl, mouse, and marmoset retina to different types of temporally varying visual stimuli. Suppression in these models is implemented via subtractive or divisive interactions of stimulus filters or by a response-driven feedback module. For the majority of ganglion cells, the subtractive and divisive models perform similarly and outperform the feedback model as well as a linear-nonlinear (LN) model with no suppression. Comparison between the subtractive and the divisive model depends on cell type, species, and stimulus components, with the divisive model generalizing best across temporal stimulus frequencies and visual contrast and the subtractive model capturing in particular responses for slow temporal stimulus dynamics and for slow axolotl cells. Overall, we conclude that the divisive and subtractive models are well suited for capturing interactions of excitation and suppression in ganglion cells and perform best for different temporal regimes of these interactions.

**Data availability statement:** The data and source code for model fitting and generating figures of this manuscript have been made publicly available at the GRO.data repository, DOI: https://doi.org/10.25625/AP5BRQ.

**Funding:** This work was supported by the European Research Council (ERC) under the European Union's Horizon 2020 research and innovation programme (grant agreement number 724822; TG) and by the Deutsche Forschungsgemeinschaft (DFG, German Research Foundation) – project numbers 432680300 (SFB 1456, project B05; TG) and 515774656 (TG). The funders had no role in study design, data collection and analysis, decision to publish, or preparation of the manuscript.

**Competing interests:** The authors have declared that no competing interests exist.

## Author summary

The spiking activity of neurons throughout the nervous system is shaped by how excitatory signals interact with mechanisms of suppression, such as input from inhibitory neurons or synaptic fatigue. For example, suppressive signals can influence a neuron's sensitivity to different sensory stimuli, the temporal structure of its activity, or how it adapts its responsiveness under different ranges of input intensity. Conceptually, one often distinguishes whether the suppression acts on the excitatory signal in a subtractive fashion, in a divisive fashion, or as a feedback signal whose strength depends on the previously elicited activity. Here, we investigated three corresponding computational models with similar overall structure but different suppressive interactions. To evaluate the models in the context of visual stimulus encoding, we assessed their ability to predict measured light responses of retinal ganglion cells, the output neurons of the vertebrate retina, in different species (axolotl, mouse, marmoset). We found that including suppression was generally better than not including suppression and that the subtractive and divisive models overall outperformed the feedback model. Moreover, the former two models were best suited for different conditions, with divisive suppression excelling for rapid stimulus kinetics and subtractive suppression better capturing slow response modulations.

## Introduction

Sensory systems dynamically encode stimuli with a wide range of temporal characteristics, reflecting the complexity of changing scenes and self-motion in the natural world. A crucial mechanism for shaping and supporting the encoding of dynamic stimuli is thought to lie in the interplay of excitation and suppression, with the latter encompassing inhibitory synaptic signals as well as cell-intrinsic suppression and negative feedback, such as mediated by voltage-dependent conductances. A simple example of this interplay of excitation and suppression is the shortening of response duration, leading to markedly transient responses of cells in the retina [1], the thalamic lateral-geniculate nucleus [2], as well as in auditory [3] and somatosensory [4,5] systems in response to an abrupt change in the strength of a stimulus. In many cases, this response transiency has been explained by suppressive inputs coming with a short delay relative to the excitatory input, resulting in a brief window of opportunity for elevated activity before suppression dominates.

Similarly, the context dependence of stimulus encoding in the visual system, such as found by varying background illumination, contrast, stimulus size, or stimulus motion, has been captured by suppressive interactions that can act as a divisive gain modification [6–8]. Suppressive mechanisms may also allow for creating dynamic variations of stimulus integration time, as demonstrated by a conductance-based encoding model of suppression [9,10]. This provides a long temporal window for integrating sensory inputs when the signal-to-noise ratio is low, letting a neuron average out noise and generate a robust response. A short integration window at higher

signal-to-noise ratio, on the other hand, enables the neuron to lock its response to the fast fluctuations of the sensory input.

To understand the role of the excitation–suppression interplay in generating various response dynamics, several filter-based encoding models have been proposed. The canonical idea behind these groups of models is to capture properties of excitatory and suppressive signals in separate temporal filters, whose signals are then combined to yield the activity of the modeled neuron. Here we focus on three classes of these models that have been proposed to capture the temporal response properties of retinal ganglion cells, using subtractive suppression [11], divisive suppression [1], and activity-dependent negative feedback [12–15], respectively. For the subtractive and divisive models, the suppression is derived in a feedforward way by applying a filter to the same input signal that feeds the excitation. For the feedback model, on the other hand, suppression is triggered by the spikes that are stochastically generated by the model and that provide the input to the suppressive filter, an approach that has been popularized in the context of the generalized linear model (GLM) [12,14,15].

To maximize the similarity between the training algorithms and optimization constraints across the different models, we trained and tested the three suppressive models, as well as a linear-nonlinear (LN) model for comparison, within a unified computational structure, where each of the models could be implemented by fixing certain model parameters to specific values. We applied this framework to ganglion cell responses recorded from isolated retinas of axolotl, mouse, and common marmoset under stimulation with full-field white-noise flicker of light intensity. The unified implementation and datasets allowed us to compare the models directly and to identify the cell and stimulus types for which each model outperforms the others.

One of the main challenges for developing encoding models is their generalization to different classes of stimuli. While we train the models on data from white-noise stimuli, as is customarily done for these and comparable models, we also test the performance of the suppressive models on stimuli that contain sinusoidal frequency and contrast sweeps. Assessing the ability of a model to generalize to such non-stationary stimuli helps provide an understanding of how the model captures retinal stimulus processing in a complex setting with varying stimulus statistics.

## Results

### Comparison of three suppression models in a unified framework

We implemented three filter-based neuron models of suppression, based on subtractive, divisive, and feedback interactions, respectively, as well as an LN model in a unified structure (S1 Fig; see Methods). The common structure aimed at maximizing similarities in the models' components as well as in training and testing procedures. In this framework, each model yields an integrated signal that encompasses the effects of excitation and suppression and that—after nonlinear rectification—is used for the stochastic generation of time-varying spike counts in small discrete time windows via a Poisson process. What differs between the models is how the integrated signal is obtained (Fig 1A). The subtractive and divisive models both had two upstream branches, each with its own stimulus filter and subsequent nonlinear transformation. The two branches were then combined by a subtractive or divisive operation, respectively. The upstream filters were unconstrained, whereas the shapes of the nonlinear transformations were restricted to support the excitatory or suppressive action: The nonlinearities for the excitatory and suppressive branches of the subtractive model as well as for the excitatory branch of the divisive model were required to be monotonically increasing, whereas the suppressive branch of the divisive model had a nonlinearity that was "bump-shaped", i.e., monotonically increasing on the side of negative inputs and monotonically decreasing on the positive side. These constraints on the nonlinearities of the suppressive branches are in line with the original formulations of the models [1,11] and ensure that the suppressive branches indeed retain their interpretation as suppressive components during model optimization, as otherwise they might contribute in an excitatory fashion (see Methods for additional explanations). The constraints are thus key ingredients of the models. The feedback model and the LN model each had a single (excitatory) upstream branch for stimulus filtering and a subsequent

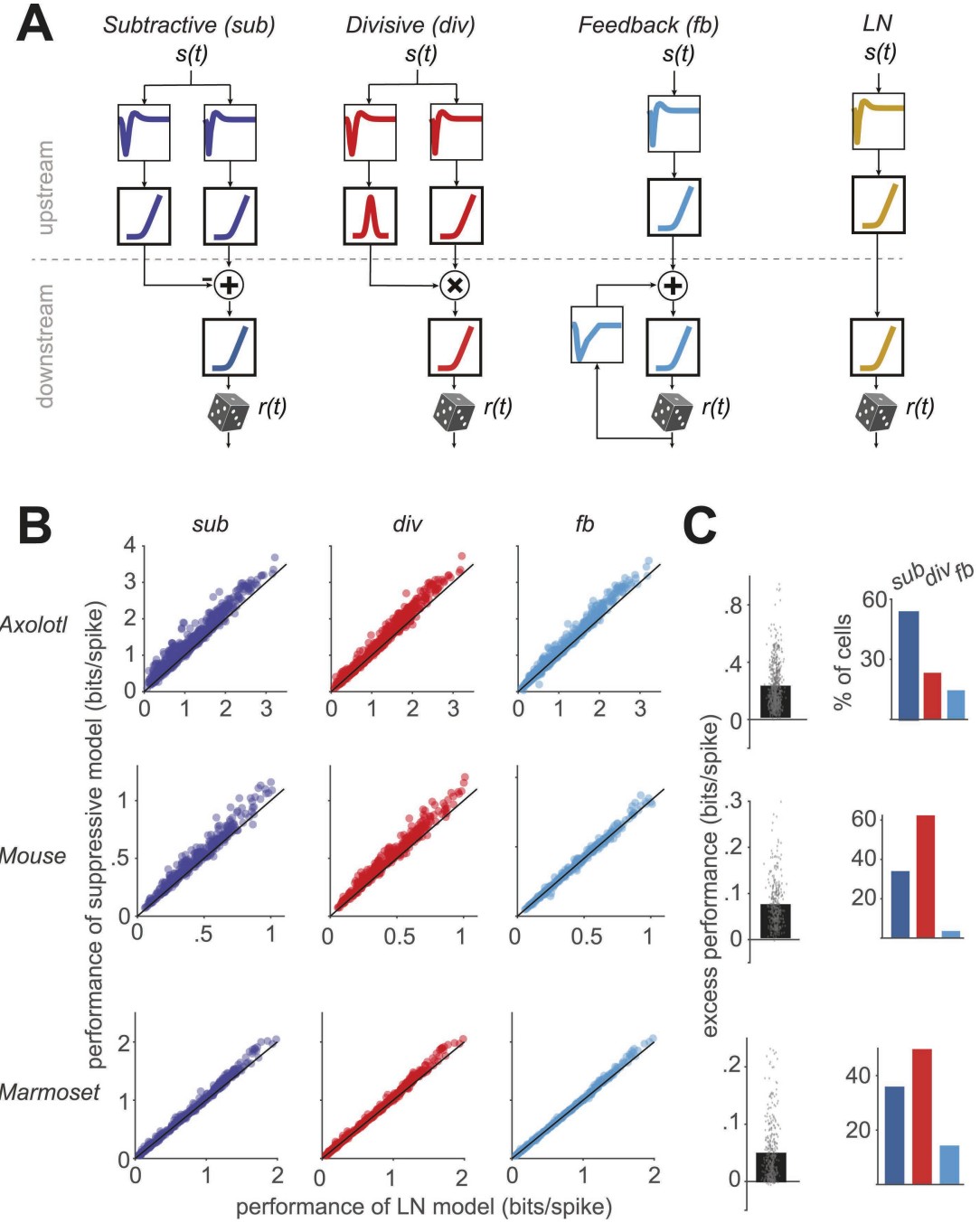

**Fig 1. Schematics and overall performance of the models. A)** Structure of the three suppressive models used in our study with subtractive, divisive, and feedback suppression and of the LN model. The filter components are shown with thin frames and the nonlinearities with thick frames. **B)** Comparison of the performance of each model, measured as the information per spike, to the performance of the LN model. Each data point represents one cell. **C)** Left: The excess performance (performance of suppressive model minus LN-model performance) of the best-performing suppressive model, determined for each cell. Right: The percentage of cells for which the corresponding suppressive model outperformed all the other models.

monotonically increasing nonlinear transformation of the filtered signal. For the feedback model, the suppression was implemented by a filter that was applied to the model-generated spikes, as introduced previously [12,14], and the upstream and feedback filter signals were then summed, similar to a GLM.

The final output nonlinearity just prior to the stochastic spike generation is a common element in all models. For the LN model, this means two successive nonlinear stages that together constitute the typical single output nonlinearity of this model. That this is here subdivided into two stages is a concession to the unified structure of all models. We checked that the two-stage nonlinearity did not impair model performance by comparing this LN model to a more conventional LN model where the filter was obtained either as the spike-triggered average or as the first stimulus feature from a spike-triggered-covariance analysis (selecting from these two options the one that performed better). The nonlinearity was here obtained as a histogram of the measured spike probability depending on the filtered stimulus signal. We found that the LN model that was fitted via the unified structure of S1 Fig, including the two-stage nonlinearity, slightly outperformed the classical LN-model implementation (S2 Fig), presumably because the parametrization and performance-based parameter optimization of the fitted model supported model performance for held-out data. For all further comparison, we therefore used the fitted LN model, according to the structure of S1 Fig.

To limit the number of free parameters, the upstream nonlinearities were parametrized using sets of basis functions and the output rectifier was a parametrized softplus function (see Methods). All models were trained using a maximum-likelihood approach with a block-coordinate ascent algorithm, in which we iterated the sequential update of each block of the model (a filter or a nonlinearity) to maximize the likelihood function while keeping other blocks constant.

The models were trained and tested on ganglion cell spiking activity in retinas of axolotls (n = 15 recordings), mice (n = 8), and marmosets (n = 10). Spikes were recorded with multi-electrode arrays while the retinas were stimulated with full-field illumination. For model fitting, we applied light intensity that varied according to Gaussian white noise. For axolotl and mouse, the stimulus consisted of an alternating sequence of non-repeating white noise, which was used for training the models, and "frozen noise", i.e., a fixed white-noise segment that was presented repeatedly, which we held out for testing (see Methods). In the marmoset data, no frozen-noise stimulus was used. Therefore, we segmented the data into pseudo-trials of 33.3 s duration, using one fifth of each trial as held-out test data and training the models on the rest. The model performance was assessed by calculating the average information per spike provided by the model about the cell's spiking response [2,16], assessed on the held-out test data segments. Note that this measure of model performance does not require repeated trials.

Our analysis included 607 axolotl, 335 mouse, and 370 marmoset cells that responded reliably to the white-noise stimulation. ON-OFF cells, i.e., cells that responded to both step-wise increases and decreases in light intensity, were removed (see Methods) because their responses cannot be captured by the monotonically increasing excitatory nonlinearities, as applied in all four models, and typically require already two parallel filters for their activating, excitatory input [17,18].

Comparing the performance of the three suppressive models to that of the LN model (Fig 1B) showed that each of the obtained suppressive models predicted responses on held-out data better than the LN model for the majority of cells (subtractive: 97% of the cells, divisive: 96%, feedback: 85%). To quantify for each cell and each suppressive model how much predictive power was gained by the addition of the considered suppressive component, we computed the excess performance as the difference (in bits/spike) between the model performance of the considered suppressive model and that of the LN model (Fig 1C). We found that, for 95% of the cells with the highest excess performance within each species, the excess performance of the best-performing suppressive model was in the range of [0.03, 0.94] bits/spike for axolotl, [0.01, 0.29] for mouse, and [0.001, 0.23] for marmoset. The best-performing model was the subtractive model for 46%, the divisive model for 42%, and the feedback model for 12% of the cells.

For recordings from axolotl and mouse retina, the repeated presentation of the fixed frozen-noise sequence (which was not used for parameter fitting) allowed us to obtain firing-rate profiles by averaging the number of spikes per time bin

over stimulus repeats. Comparison of measured firing rates with model-predicted firing rates then provided a measure of explained variance (S3 Fig). This yielded qualitatively similar findings as the evaluation via per-spike information (cf. Fig 1B and 1C). In particular, the explained variance was generally higher for all suppressive models than for the LN model (p << 0.001 in all three cases; Wilcoxon signed-rank test). Moreover, the excess performance, as obtained here by subtracting the explained variance of the LN model from that of the considered suppressive model, was generally larger for the subtractive and the divisive models as compared to the feedback model, corroborating that the subtractive and divisive models on average outperformed the feedback model.

## Cell-type specific comparison of subtractive and divisive model

As the subtractive and divisive models generally had the top performance among the four assessed models, we compared these two models more closely and aimed at determining for which cells one or the other of these two models was superior. Comparing the performances on a cell-by-cell basis (Fig 2A) indicated that, in the axolotl retina, there is a group of cells for which the subtractive model outperformed the divisive model. For the rest of the axolotl cells as well as for mouse and marmoset cells, the performances of the two models were overall comparable. For mouse and marmoset, the divisive model performed overall slightly better than the subtractive cells (p<0.003 in both cases; here and in later comparisons between models: Wilcoxon signed-rank test with false discovery rate for multiple comparison correction [19]; Fig 2A).

For the axolotl retina, we aimed at understanding the diversity in model performance by classifying the cells into ON- and OFF-type classes as well as into slow and fast cells. To do so, we computed the temporal filter for each cell as the spike-triggered average under the temporal white-noise stimulation and clustered cells in the space of the first two principal components of all filters into four clusters (S4 Fig), which were characterized according to the sign of the primary peak of the filter as ON- or OFF-type and according to the peak latency as fast or slow. Based on this classification, we found that the relative performance of the subtractive and the divisive model depended on the kinetics of the considered cell (Fig 2B). For fast cells, the difference between the performance of the subtractive and the divisive model was small, with the subtractive model just slightly better for fast OFF cells (mean difference=0.026 bits/spike, p=0.005) and no significant difference for the small number of identified fast ON cells (mean difference=0.02 bits/spike, p=0.7). For slow ON and slow OFF cells, on the other hand, the subtractive model performed substantially better than the divisive model (slow OFF: mean difference=0.08 bits/spike, p << 0.0001, slow ON: mean difference=0.1 bits/spike, p << 0.0001).

To better understand how the suppressive components of the subtractive and the divisive model help improve the models over the LN model, we examined the model components for individual sample cells more closely. For the sample axolotl slow OFF cell in Fig 3A, the subtractive and the divisive model yielded qualitatively similar stimulus filters for both the excitatory and the suppressive model branches (Fig 3Ai). Yet, the influence of the suppressive branch on the predicted activity is markedly different, in particular during time stretches where the output of the excitatory filter branch is small or subthreshold (Fig 3Aii). For the subtractive model, the output of the suppressive branch continues to be modulated by the stimulus even when the excitatory branch yields a flat (non-zero) output and thereby allows the model to capture moderately-sized spiking events during these periods (arrows in Fig 3Aii). For these responses, the model activity essentially occurs through release from the suppressive input, akin to a disinhibition mechanism. For the divisive model, on the other hand, even though its suppressive component also displays similar modulations (with more positive values here corresponding to a reduction in the suppression), the response prediction does not capture the cell's firing rate peaks when the excitatory signal remains constant. This makes intuitive sense because the excitatory signal is low (or even zero) during these periods, and a divisive mechanism, unlike the subtractive mechanism, cannot substantially modulate an excitatory signal that is small to begin with and even has no effect at all when the excitatory signal is exactly zero.

In addition, the shape of the nonlinearity in the suppressive branch also differs between the two models. In the subtractive model, the monotonic shape of the nonlinearity in the suppressive branch allowed for a gradual decline of suppression as the activation of the suppressive filter becomes more negative (Fig 3Ai). This supports the gradual and nuanced

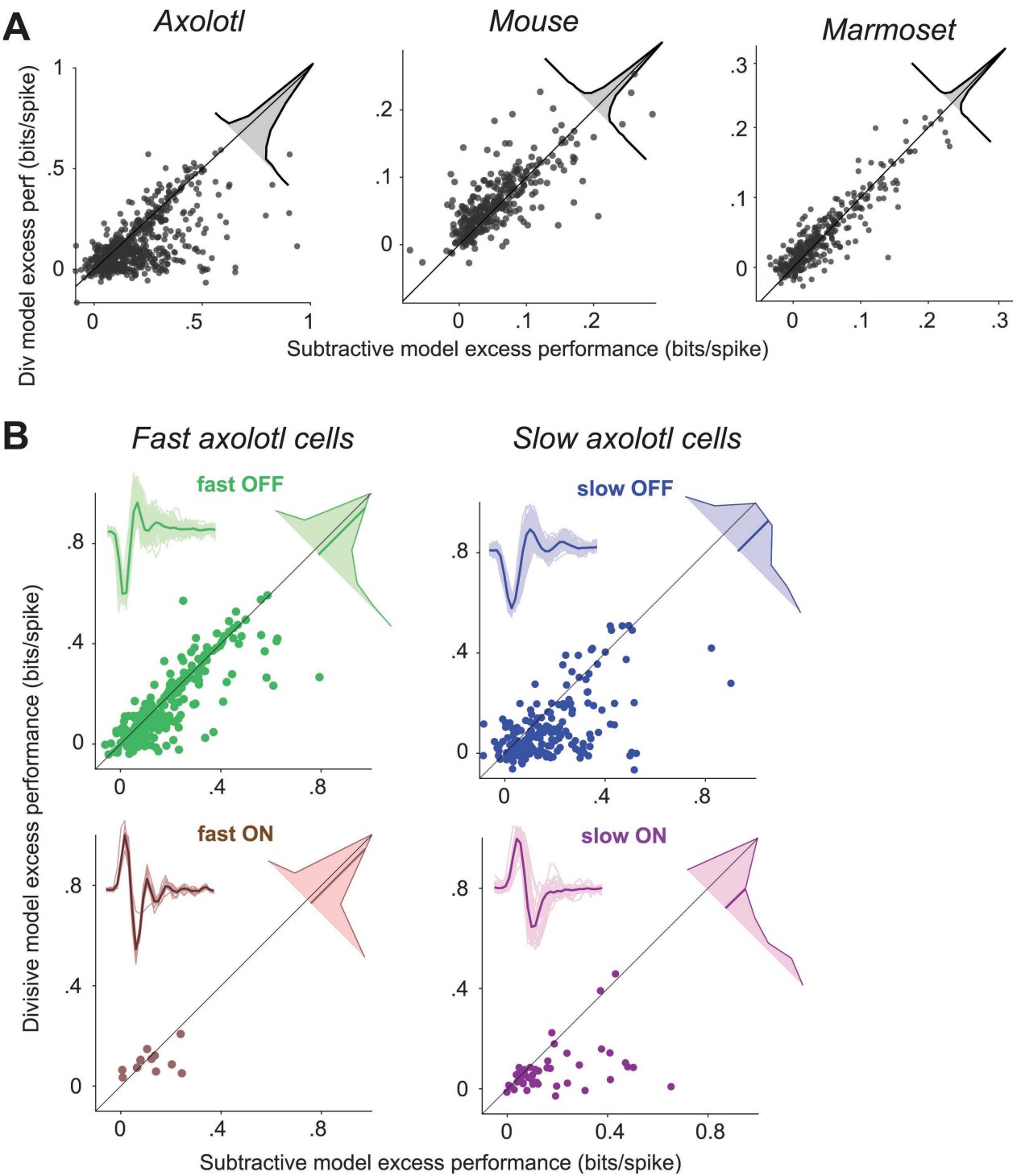

**Fig 2. Comparison of the excess performance of the subtractive and the divisive model. A)** Scatter plot of the excess performance of the subtractive model vs. the divisive model for each species. Each data point is a cell. Insets: The distribution of the performance difference between the subtractive and the divisive model. **B)** Same as A, but for slow/fast and ON/OFF subgroups of the cells recorded from axolotl retina. Insets: The filters of the LN model of all cells within the subgroup (thin lines) and their average (thick line).

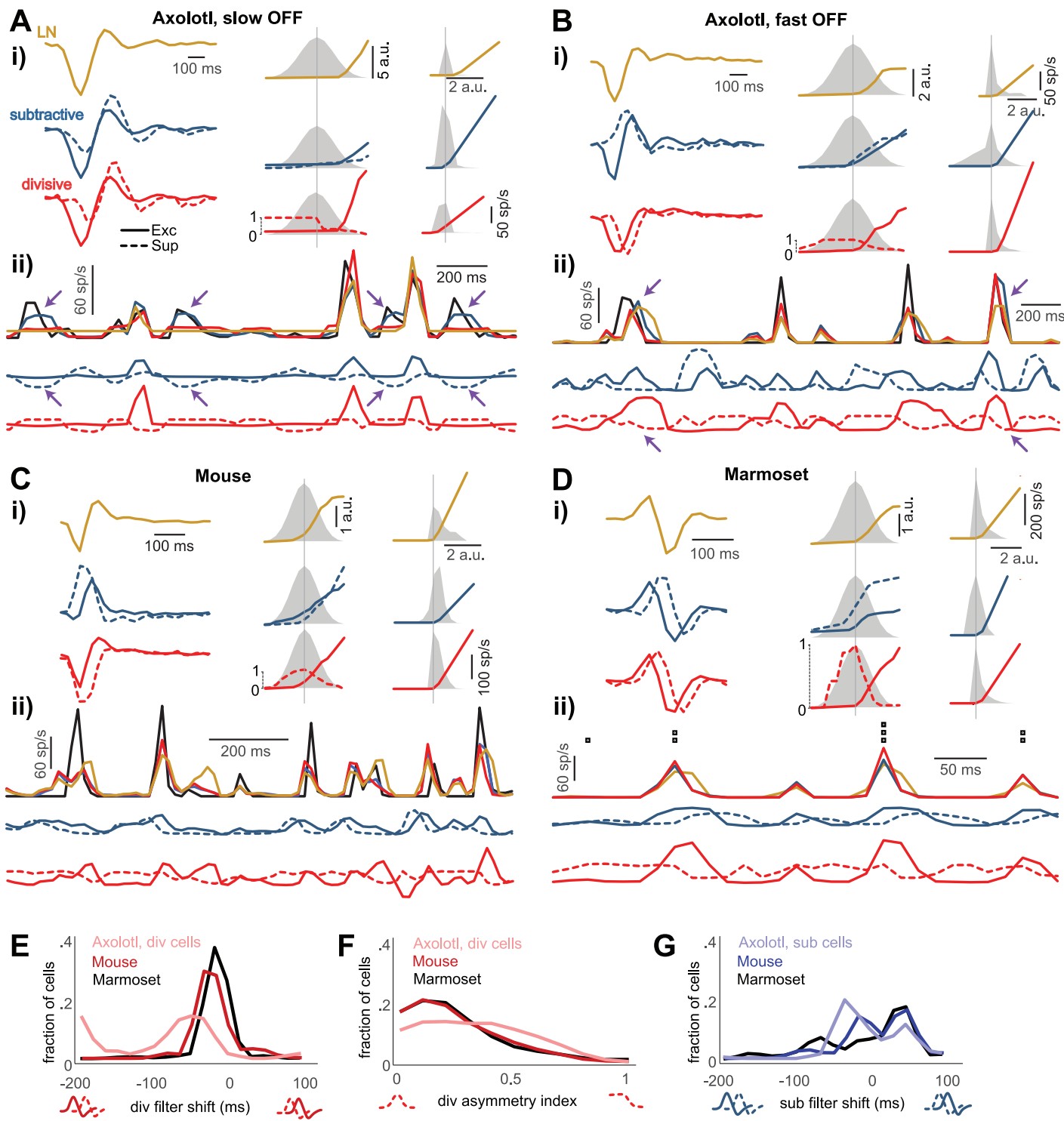

**Fig 3. Model fits and response predictions for sample cells from each investigated species. Ai)** Excitatory and suppressive filters (left), upstream nonlinear functions (middle), together with the distribution of the corresponding input (gray), and output rectifier (right), together with the distribution of the corresponding input (gray). The feedback model was omitted, due to its lower performance, but the LN model was included for comparison. The y-axis scale of the upstream nonlinearities is marked by the range zero to unity for the divisive model (corresponding to the range of the suppressive output) and by arbitrary units (a.u.) otherwise, which correspond to the a.u. on the x-axis of the output rectifier. **Aii)** For a subsection of the test-trial period:

Top: The cell's firing rate (black) and the predictions of the three models in spikes per second (sp/s). Below: The output of the excitatory (solid line) and suppressive (dashed line) upstream branches (taken after the upstream nonlinearities) for the subtractive (blue) and the divisive model (red). Arrows here mark spiking events captured by the subtractive model but not by the other two models as well as the corresponding decrease in suppression of the subtractive model. **B, C, D)** Same as A for three further sample cells. Note that D contains no measured firing rate because the marmoset recordings did not contain frozen-noise sections, but instead shows the single-trial spike train above the prediction curves, with every black square denoting a spike in the corresponding time bin. Arrows in B mark spiking events where the divisive model was most successful in correctly curbing the tail end of the response prediction as well as the corresponding suppression signals of the divisive model (with smaller values here denoting stronger suppression). **E)** The shift in the suppressive filter of the divisive model, relative to the excitatory filter, across cells. The shift was defined as the time lag for which the cross-correlogram between the excitatory and suppressive filters obtained its maximum. **F)** The asymmetry index for the suppressive nonlinear function of the divisive model. The index was calculated as the absolute difference between the sum of the function values over negative and positive inputs, respectively, divided by the total sum of the function values. **G)** The shift in the suppressive filter of the subtractive model, relative to the excitatory filter across cells, calculated as in E.

contribution of suppression to the model activation (Fig 3Aii). In the divisive model, on the other hand, the shape of the nonlinearity in the suppressive branch led to a unilateral, rectified suppression with essentially no sensitivity to filter activations near to or smaller than zero (Fig 3Ai). Note that this shape is partly influenced by the constraint on this nonlinearity to be bump-like so that suppression cannot decrease as the filter activation deviates from zero. This is, in fact, a fundamental constraint of the divisive model to ensure that the corresponding branch indeed act as a suppressive mechanism (see Methods), explaining further why the divisive model might miss some of the moderately-sized response peaks (Fig 3Aii).

By contrast, for the sample axolotl fast OFF cell in Fig 3B, the divisive model captured the shape of the response peaks slightly better than the subtractive model, in particular by earlier curbing the final part of some response events (arrows in Fig 3Bii). At these time points, the suppressive branch yields a low, near-zero output (arrows in the bottom row), which, via the multiplication that constitutes the divisive interaction, is effective at suppressing the strong signal supplied by the excitatory branch. The response-peak curbing is further aided by the fact that the filter of the suppressive branch came out to be similar in shape to the excitatory filter, but with a temporal shift, thus providing suppression with a slight delay to the excitatory input.

Suppressive filters in the divisive model that are approximately temporally delayed versions of the excitatory filters are also found for the sample mouse cell shown in Fig 3C and the sample marmoset cell in Fig 3D. As for the sample axolotl fast OFF cells, this resulted in a suppressive response that came with a slight delay compared to the excitatory response, leading to a timelier prediction of the falling edge of firing-rate response peaks compared to the LN model.

This delayed suppression in the divisive model was a general observation. Often, the suppressive filters lagged the associated excitatory filters by 1–2 stimulus frames (around 30–70 ms for axolotl and around 20–30 ms for mouse and marmoset). This was the case for 57% of the axolotl cells among those for which the divisive model outperformed the others, 79% of mouse cells, and 70% of marmoset cells, as assessed by the peak location in the cross-correlogram between the two filters (Fig 3E). Also, for 57% of the axolotl cells with a superior divisive model, 90% of the mouse cells, and 88% of the marmoset cells, the divisive nonlinearity was fairly symmetric (Fig 3F; asymmetry index < 0.5), as measured by the absolute difference in average values of the nonlinearity on the left versus right side of zero (normalized by the corresponding sum). Thus, for these cells, both positive and negative activations of the suppressive filter led to suppression in the model, in contrast to the example of the slow axolotl cell in Fig 3A.

The subtractive model for the three sample cells shown in Fig 3B and 3C, on the other hand, yielded a more diverse set of suppressive filters, relative to their excitatory counterparts. The suppressive filter could be temporally trailing the excitatory filter (e.g., Fig 3D), preceding it (Fig 3C), or have an altogether different shape or relative strength in the positive and negative components (Fig 3B). A suppressive filter lagging the excitatory one by 1–2 stimulus frames was found for 40% of the axolotl cells for which the subtractive model outperformed the others, for 12% of mouse cells, and for 5% of marmoset cells. By contrast, 32% of those axolotl cells, 23% of the mouse cells, and 28% of the marmoset cells had suppressive filters leading the excitatory filter by 1–3 stimulus frames (Fig 3G). Yet, model performance was generally similar to

the divisive model, suggesting that different computational motifs might converge to inherently different, but equally good solutions.

### Differences in model performance between sustained and transient cells

Delayed suppression that comes with a similar stimulus dependence as excitation is a proposed mechanism for generating sharp, transient responses [1,2]. The delayed suppression may result, for example, from feedforward inhibition with additional synaptic transmission delays as compared to excitation, which generates a brief window for excitation to elicit spikes before the suppressive component terminates the response.

   We thus hypothesized that cells with transient response characteristics have particularly strong suppression and that adding a suppressive component in the model is especially important for these cells. For a subset of marmoset cells, which had also been stimulated using full-field steps in light intensity (100% contrast), we therefore determined the transiency of their responses to light-intensity steps and classified them as either transient or sustained by comparing the strength of the initial response peak to the sustained activity beyond 300 ms after stimulus onset (Fig 4A). We then checked for each class of cells how important the addition of a suppressive component in the model was by collecting the excess performance values. As the model performances displayed considerable variability across experiments, most likely at least partly due to different sampling of ganglion cell types and retinal eccentricity, we made this comparison on an experiment-by-experiment basis. Furthermore, we focused on ON-type cells, as some of our recordings did not yield a sufficient number of OFF-type cells for separation into transient and sustained classes.

   For the sample recording shown in Fig 4B, the excess performance of suppression (from the best-performing suppressive model for each cell) was significantly larger for the transient ON cells compared to the sustained ON cells, suggesting weaker suppression in the sustained ON cells (mean difference: 0.032 bits/spike; p=0.0083). Four out of six recordings showed a similar difference between the transient and sustained ON cells, whereas there were smaller or no substantial differences for two recordings (Fig 4C). This indicates that, indeed, explicit suppressive model components can be particularly helpful for transient cells in explaining response characteristics.

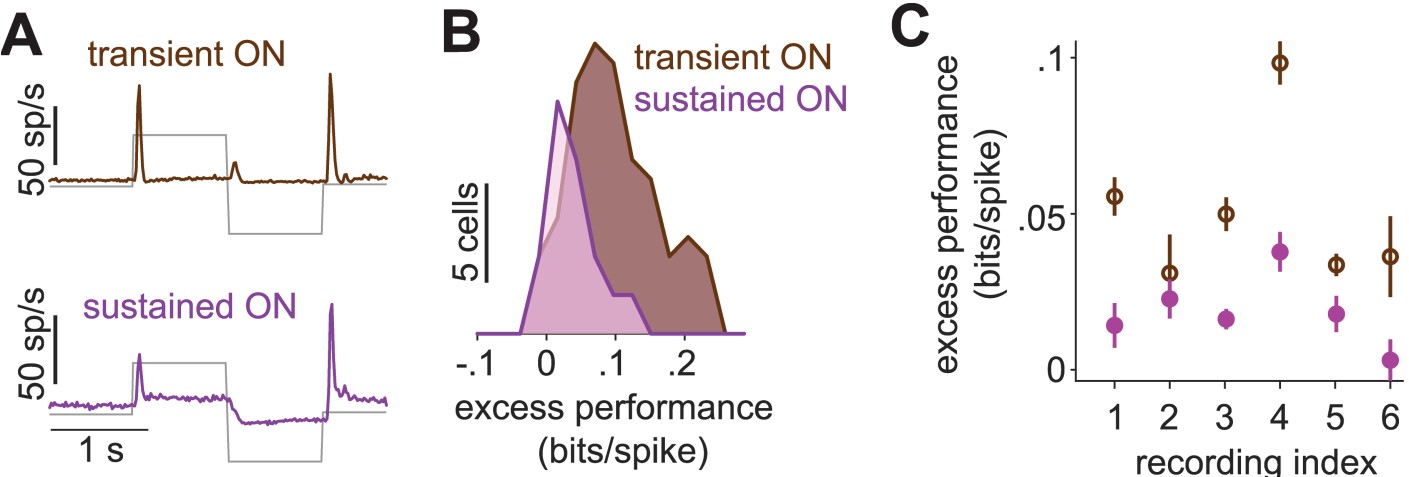

**Fig 4. Suppression for transient versus sustained marmoset cells. A)** Responses in spikes per second (sp/s) of marmoset cells to a step stimulus for transient ON and sustained ON cells, averaged across cells in a sample recording (recording index=4). **B)** Distributions of the excess performance of the best-performing model, separately for transient and sustained ON cells for the same sample recording. **C)** Mean and standard error of the excess performance for transient and sustained ON cells in each recording.

## Generalization to stimuli with varying contrast and temporal frequency

The presented findings so far were mostly based on the performance of the suppressive models on white-noise stimuli. Thus, an important question is how the derived models generalize to stimuli with different, non-stationary statistics, such as variations in contrast or temporal frequency. For a subset of recordings from marmoset cells, we applied a full-field chirp stimulus [20], which contained two sections of sinusoidal light-intensity modulations, one at 4 Hz with monotonically increasing contrast from 0% to 100% over 8 s ("contrast sweep") and the other at 100% contrast with increasing frequency from 0 Hz to 15 Hz over 8 s ("frequency sweep"). We used measured firing rates in response to repeated presentations of these two sinusoidal light-intensity modulations to test the performance of the models after training on the white-noise stimulus.

As illustrated by the model responses for two sample cells shown in Fig 5A-D, the degree to which the prediction of each model matched a cell's response varied with the frequency and contrast of the stimulus. For the sample cell shown for the contrast sweep (Fig 5A), all models captured the response profile at low contrast quite well, displaying rectified response peaks with increasing amplitude for increasing contrast, which matched the recorded activity. For higher contrast, however, most models gave overshooting predictions, with response peaks that were too large and too long in time (Fig 5A, right). Only the divisive model managed to yield good predictions here, reflecting how the cell's response events at high contrast were cut off and suppressed after the initial rising phase.

We quantified the dependence of model performance on contrast for the contrast sweep stimulus by calculating the explained variance ($R^2$) for each model within a sliding window of 150 ms duration, shifted along the time course of the stimulus in steps of 16.7 ms (1 stimulus frame). For the sample cell, the divisive model substantially outperformed the others for any contrast higher than about 20% (Fig 5B). To quantify the performance across the population of cells, we calculated the performance range for each cell as the range of contrasts for which $R^2 > 0$. Over the population, the performance range was significantly larger for the divisive model compared to other models ($p \ll 0.0001$ in each case; Fig 5E), indicating that the divisive model generalized over a wider range of contrasts. This finding resonates well with a previous finding in mouse ON-alpha ganglion cells [1], which had shown that the divisive model can capture adapting responses under contrast changes in a white-noise stimulus.

Similar to the contrast dependence of model performance, we observed that the relative performance of the models depended on temporal frequency. For the sample cell shown for the contrast sweep (Fig 5C), model predictions under low-frequency intensity modulations were often too wide and shallow. Here, the subtractive model managed best to generate fairly sharp responses, approximately matching the time course of the data. For larger temporal stimulus frequencies, on the other hand, all models tended to overestimate the duration of response events (Fig 5C, right). Yet, similar to the high-contrast scenario in Fig 5A, the divisive model managed best to cut off some of the response prediction after the initial rising phase and thus yielded slightly more accurate predictions at higher frequencies.

We again quantified the frequency dependence of model predictions by computing $R^2$ within a sliding window, capturing different frequency ranges. As shown in Fig 5D, the subtractive model outperformed the other models for the sample cell when the frequency of the stimulus was below about 4 Hz, whereas the divisive model had the highest performance among the tested models for higher frequencies. To assess the generality of these findings at the population level, we calculated for each cell the low-frequency performance, i.e., the average explained variance for the first 300 ms of the frequency sweep (corresponding to 10 shifts by one stimulus frame of the 150-ms analysis window) as well as the performance range, i.e., the range of frequencies for which $R^2 > 0$. For the subtractive model, the low-frequency performance was indeed significantly higher than for the other models ($p < 0.03$ for each comparison; Fig 5F, left), indicating that this model performed best in capturing the cells' responses to low frequencies in the chirp stimulus. On the other hand, it was again the divisive model—like for the contrast sweep—that had the largest performance range ($p < 0.004$ for each comparison; Fig 5F, right), indicating that this model generalized better over the frequency range of the chirp stimulus.

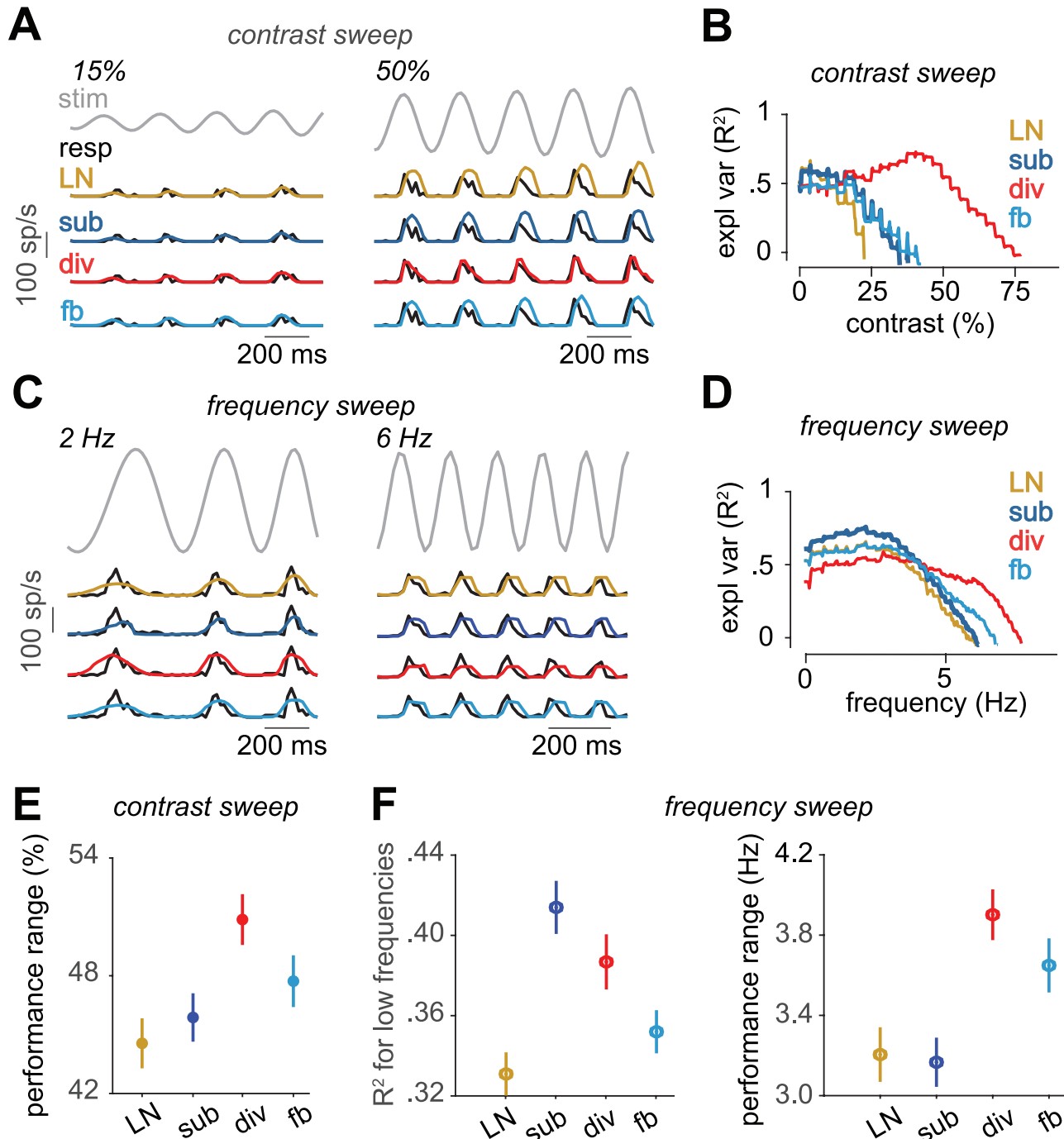

**Fig 5. Evaluation of the models on responses to contrast and frequency sweeps for marmoset cells. A)** Part of the contrast-sweep stimulus together with the response of a sample cell and the model predictions in spikes per second (sp/s). Two sections of the contrast sweep, with indicated starting contrast and a duration of about 4-5 cycles, are shown. The complete range of the stimulus, the cell response, and the model predictions are provided in S5 Fig. **B)** Explained variance of the models for the cell in A, computed within a 150 ms sliding window. The x-axis shows the starting contrast of the sliding window. For each model, only the range with $R^2 > 0$ is shown. **C)** and **D)** Same as A and B, but for the frequency sweep and for a different sample cell. **E)** The contrast range for which each model had $R^2 > 0$ (mean and standard error over cells). **F)** Left: The $R^2$ of each model averaged over the first 10 data points of the frequency sweep analysis (mean and standard error over cells). Right: The frequency range for which each model had $R^2 > 0$ (mean and standard error over cells).

## Discussion

The interplay of excitation and suppression is a central theme for the generation and formation of neuronal responses to sensory stimulation. How to reflect the suppression in computational models of single cells is thus a question of broad interest. By testing models on spike-train data from retinal ganglion cells of three different species, we found that three previously introduced approaches to incorporate suppression into filter-based models can all be used to improve response predictions over the commonly applied single-filter model, the linear-nonlinear model. Moreover, the subtractive and the divisive model generally outperformed the feedback model for most cells in axolotl, mouse, and marmoset.

### Comparisons of model performances

The fact that the subtractive and the divisive model overall performed quite similarly for most cells might seem a bit disappointing at first glance, as the lack of a clear "winner" makes inference about the type of excitation–suppression interaction difficult. On the other hand, this null-result means that, from a phenomenological perspective, the details of how suppression is included in the model matter much less than having a suppressive component at all, indicating that different interactions can achieve similar functions by different means [21]. The functional equivalence of different model structures is reminiscent of studies of the stomatogastric ganglion in lobster where the potential variable implementations of the same circuit function has been forcefully demonstrated [22].

More detailed comparisons between these two models, however, revealed systematic differences in how performance depended on species and stimulus features. The subtractive model performed better for capturing responses of slow axolotl cells to white-noise stimuli and responses of marmoset cells to low-temporal-frequency parts of the chirp stimulus. The divisive model, on the other hand, displayed greater potential for generalizing over wider ranges of contrast and temporal frequency. This suggests that different neural suppression mechanisms may dominate for different temporal scales of stimulation and response and that the two models differ in how well they capture the effects of these specific mechanisms. In particular, the subtractive model may specifically account for biological mechanisms that prevail in slow cells or that take effect when the stimulus is dominated by slow frequencies.

A technical aspect that may contribute to limiting the performance of the feedback model is the fact that suppression in this model depends on the occurrence of spikes, which are sparse and stochastic in the model. This limits the effectiveness of the suppression mechanism, e.g., it cannot suppress a spiking event that is not preceded by another spiking event. Alternative implementations might let the feedback depend on the continuous internal signal before spike generation [23] or might apply a deterministic spike generation process [13,24].

### Relation to previous work

Multi-filter suppression models have been applied to describe response characteristics of retinal ganglion cells and other neural systems in various studies, though rarely with a direct performance comparison of multiple models or across species. The divisive model has been shown, for example, to accurately capture the temporal precision and contrast adaptation effects for a specific type of mouse retinal ganglion cells [1]. Our findings indicate that this may hold more generally across cell types and species. Similarly, feedback models with feedback based on elicited spikes [13,24] can help improve prediction of spike timing over comparable single-filter models, mostly owing to providing for a refractory period [25]. Previous work had suggested that combining divisive suppression and an explicit refractory period may further improve predictions of response reliability over models with only one of these mechanisms [1], yet with divisive suppression providing the larger contribution for the studied mouse cell type.

Our implementation of feedback suppression was similar to the popular GLM approach [12,14], where the suppression is triggered by the generated spikes. Other filter models have incorporated feedback suppression by using the output of the primary, activating filter as the source signal for the feedback filter before spike generation [23]. This has been successfully used to capture the dynamics of retinal ganglion cells to motion stimuli, including the early, "anticipatory"

responses to moving versus flashed stimuli [23], enhanced responses to motion onsets [26] and reversals [27], and the population-based tracking of moving objects [28]. Interestingly, this feedback model, which has also been called LfN model, owing to the insertion of the feedback between the linear and nonlinear components of the LN model [28], is structurally more similar to our divisive model than to our feedback model. The reason for this is that the feedback component in the LfN model is obtained from the primary, excitatory pathway by an additional filtering process, typically using a decaying exponential in time, with no stochastic spiking mechanism in between, and a nonlinear transformation. Thus, the feedback component can also be viewed as a deterministic feedforward signal that is obtained through a concatenation of the primary filter and a subsequent nonlinear filtering transformation. In the LfN model, this signal then multiplicatively modulates the output of the primary filter, thus making the interaction of the LfN model's feedback with the primary filter pathway similar to the interaction of the two filter pathways in our divisive model.

For the present case of stimuli with no spatial structure and purely temporal intensity modulations, the LfN model can thus be approximately viewed as a special case of a divisive model, albeit with slightly different constraints, as the suppressive filtering is approximately a low-pass-filtered version of the primary filter and the primary filter pathway itself has no independent nonlinearity, whereas the suppressive filter applies a decaying nonlinearity of predefined shape. Note, though, that our divisive model often indeed yielded suppressive filters that were consistent with low-pass-filtered or at least delayed versions of the excitatory filter (Fig 3E) and that the nonlinearity of the suppressive filter can look similar to the decaying filter employed in the LfN model (e.g., Fig 3A and 3B).

In the case of spatiotemporal stimuli, which have been the primary use case of the LfN model, the interpretation of the suppressive component in the LfN model becomes clearer, as a single spatial filter is applied to yield the primary input signal [23,26–28], which differs from the straightforward generalization of the divisive model where each filter pathway may have its own spatial filter, similar to extensions of the subtractive model [17,29]. Thus, under spatiotemporal stimuli, multi-filter pathways such as the subtractive and divisive model studied here, might aim at capturing the spatiotemporal signature of suppression in the context of nonlinear spatial processing [30], whereas the LfN model provides a data-efficient approach to model suppression with feedback acting on a spatiotemporally filtered signal [26].

### Model complexity and limitation to spatially homogeneous stimuli

In the present study, the implementation of the compared models was guided by the goal of a common framework that made the model structures and the training most comparable. Yet, the number of free parameters was not identical between all models. Given the complexity of the models, this raises the question of whether the comparison might be influenced by over-fitting, which could reduce performance on the test data for models with a higher number of degrees of freedom. However, all evaluations occurred on held-out data not used for model fitting, and the subtractive and the divisive model actually had the same number of free parameters so the observed differences are unlikely to arise from differences in fitting accuracy. Therefore, we believe that over-fitting is not likely to affect the interpretability of our findings.

A limitation of the present study is that we restricted our analysis to the use of full-field stimuli without any spatial structure. We made this choice in order to restrict the number of free parameters in the models as well as to allow for comparisons between cell types and models independent of how well the spatial filtering and potential nonlinearities in spatial stimulus integration [30–33] are captured. On the downside, the use of full-field stimuli makes our analysis insensitive to the spatiotemporal structure of suppression, in particular by not providing for a dedicated suppressive signal pathway from the receptive-field surround. Suppressive mechanisms in the receptive-field center and surround partly result from different biophysical mechanisms and may therefore differ in whether subtractive or divisive operations dominate. Besides suppression through inhibitory signals from horizontal and amacrine cells, which originate in both the receptive field center and surround, suppression in the receptive-field center can, unlike in the surround, also occur in the direct excitatory pathway via synaptic fatigue between bipolar and ganglion cells [34,35].

Yet, which biophysical mechanisms are best described by which of our suppressive models is unclear and likely dependent on stimulus context. Thus, dedicated experiments might be needed to assess whether center and surround differ in how suppression may be captured by models. Stimulating center and surround with independent white-noise sequences might allow a straightforward extension of the present analyses with distinct suppression pathways for center and surround. Alternatively, fitting such models to responses under stimulation with natural movies might also be feasible if model parametrizations are selected that do not let the number of free parameters increase dramatically over the case with no spatial structure.

Considering spatially structured stimuli also raises the question whether suppression acts globally, affecting the sensitivity of the ganglion cell throughout its receptive field, or locally, for example in receptive field subunits, that is, on the level of bipolar cells. Comprehensive models with suppression acting at the global level or on local subunits may help distinguish the relative contributions [26,30,36], but fitting such models is no simple task given the many model parameters and their nonlinear interactions. The observation that contrast adaptation can occur for the entirety of the receptive field as well as show effects confined to individual subunits in different ganglion cells [37–39] indicates that both local and global suppression components play a role. Similarly, modeling axolotl ganglion cell responses to both smooth motion and motion onsets has indicated the need to include suppressive effects both on the bipolar- and the ganglion-cell level for a comprehensive description of suppression [26].

## Comparisons across different species: general features and species-specific variations

A particular aspect of our study was the comparison of the models across ganglion cells from different species. This showed an overall high level of consistency of the findings; the similar performance of the subtractive and the divisive model and the generally lower performance of the feedback model was observed in axolotl as well as mouse and marmoset. One somewhat exceptional group of cells was the set of slow axolotl cells that displayed a clear preference for the subtractive model over the divisive model. For these cells, the slowness is particularly pronounced, as filters and response kinetics in the axolotl are overall slower than in the other species, perhaps partly owing to the lower temperature at which the axolotl experiments were performed, consistent with their lower body temperature. This may make the subtractive model, which appeared to generally capture slow temporal dynamics better, particularly suited for the slow cells in this slow retina.

For marmoset ganglion cells, we found that including suppression in the response model, either through the subtractive or the divisive model, was more relevant for transient cells than for sustained cells. In the primate retina, the ganglion cell types of midget and parasol cells numerically make up the majority of the ganglion cell population [40,41], and among these, parasol cells tend to have transient responses and midget cells sustained responses [41–43]. It thus seems likely that it is the parasol cells for which including suppression in the model is particularly relevant, though we have no independent classification of parasol and midget cells for the present datasets.

## Generalization beyond white-noise stimuli

Like many other investigations of neuronal response models to sensory stimulation, we based our model fits on recordings under stimulation with white-noise sequences of light intensity. There is increasing awareness, however, that an important follow-up step is to assess how well the obtained model descriptions and deduced insights hold in the context of different, typically more complex stimulus characteristics, such as natural stimuli. In the present study, rather than using natural stimuli, we investigated the generalization to non-white-noise, non-stationary stimuli, with variations in specific stimulus components, such as contrast and temporal frequency. Some studies of models without suppressive components have been used for different contrast levels by recalculating the stimulus components for each contrast [31]. Other investigations have suggested suppressive elements by feedback [23,44,45], by divisive gain control [46], or by complex biologically motivated combinations of nonlinear filtering and feedback [47] as mechanisms that capture contrast adaptation and

thereby promote generalization across contrast levels. These findings are in line with our observation that suppressive model elements help generalization across contrast levels and frequency ranges and that the divisive model is particularly suited for this purpose.

### Functional interpretation

Our results indicate that the subtractive and the divisive model excel in different processing regimes, the former for slow response components (slow cells as well as stimulus sections with slow dynamics), the latter for fast responses. It thus seems plausible that these two types of suppressive interactions are each best suited for different functional goals. The divisive model seems to largely act by trimming the temporal duration of response events (e.g., Figs 3B and 5) via suppressive filters that are approximately delayed versions of the excitatory filter (Fig 3E). The divisive interaction may be particularly suited for this purpose, as it allows effective truncation of events resulting from strong activation without overly suppressing weaker signals, as apparent in the responses to contrast sweeps (Fig 5A and 5B). For slow signals, on the other hand, subtractive suppression might be the more effective mechanism, as it may allow more precise shaping of the temporally extended but weaker signals surrounding or apart from the peak activity (e.g., Fig 3A). The diversity and complexity of suppressive mechanisms in actual neurons likely comprise both subtractive and divisive components, and the relative performance of the two models can thus either depend on whether a given cell tends to generally produce rapid and brief or slow and sustained responses (Fig 2B) or whether the cell responds to fast or slow stimulus components (Fig 5C and 5D).

Individual cells that preferred the subtractive over the divisive model were particularly found in the axolotl retina (Fig 2), but it remains unclear whether this reflects a species difference, a difference in the experimental sampling of the various ganglion cell types in each species, or a difference in recording conditions, e.g., room temperature for axolotl recordings versus heated bath in the other two cases. One hypothesis is that many cells in the amphibian retina often respond sparsely to visual stimuli [48], which might reflect weak excitatory inputs and thus a relative ineffectiveness of divisive interactions to affect responses compared to subtractive suppression which could then still be capable of influencing the activity, as illustrated by Fig 3A.

### Beyond the retina

Developing and evaluating filter-based models of suppression is also of interest beyond the retina. In particular, divisive normalization [6], which operates similarly to the divisive suppression used here, has been suggested as a canonical neural computation for many cortical and sub-cortical brain regions [49]. There is thus considerable interest in differentiating, for example, the contributions of subtractive and divisive components of cortical suppression [50,51]. In particular, previous work has focused on relating subtractive and divisive inhibition to different types of inhibitory interneurons across different sensory cortical areas [52–54]. Also, the distinction between subtractive and divisive suppression has been useful for investigating functional consequences of different cellular and synaptic suppression mechanisms [55,56]. Insights gained by comparing models with different types of suppressive interactions as done here may thereby help provide general insights into the various interactions of excitation and suppression and inspire related studies of suppression in different brain areas [1,11].

## Methods

### Ethics statement

All the experiments and procedures conformed to national and institutional guidelines. Recordings from axolotl and mouse retina were approved by the institutional animal care committee of the University Medical Center Göttingen (protocol number T11/35). Marmoset retinas were obtained from animals used by other researchers, as approved by the institutional animal care committee of the German Primate Center and by the responsible regional government office (Niedersächsisches Landesamt für Verbraucherschutz und Lebensmittelsicherheit, permit number 33.19-42502-04-17/2496).

## Electrophysiology

The axolotl recordings were performed using 60-electrode perforated multielectrode arrays (MultiChannel Systems, Reutlingen, Germany; 30 µm electrode diameter and 100 µm minimum electrode spacing) as described previously [57,58]. The eyes were enucleated after 1 hour of dark-adaptation, and the retina was mounted over a 1.5-2 mm hole on a nitrocellulose filter membrane, which was then placed on the array with the ganglion cells facing the electrodes. During the experiment, the retina was perfused constantly with oxygenated (95% $O_2$ and 5% $CO_2$) Ringer's medium (110 mM NaCl, 2.5 mM KCl, 1 mM $CaCl_2$, 1.6 mM $MgCl_2$, 22 mM $NaHCO_3$, 10 mM D-Glucose monohydrate) at room temperature (~22°C).

The mouse recordings were done with 252-electrode arrays (MultiChannel Systems, Reutlingen, Germany; 30 µm electrode diameter, and 100 or 200 µm minimum electrode spacing) as described previously [39,59]. The mice (aged 8–13 weeks, of either sex) were dark-adapted for at least 1 hour and subsequently euthanized by cervical dislocation. The eyes were removed quickly and transferred to a chamber with oxygenated (95% $O_2$ and 5% $CO_2$) Ames' medium (Sigma-Aldrich, Munich, Germany), buffered with 22 mM $NaHCO_3$ (to maintain a pH of 7.4) and supplemented with 6 mM D-glucose. The eyes were dissected, and the cornea, lens, and vitreous humor were carefully removed. The retina was isolated from the pigment epithelium and transferred to a multi-electrode array. The temperature of the recording chamber was kept constant at around 32–34°C, using an inline heater (PH01, MultiChannel Systems, Reutlingen, Germany) and a heating element below the array (controlled by TCX-Control 1.3.4, MultiChannel Systems), while the retina was perfused continuously with oxygenated Ames' medium (4–5 ml/min).

For both axolotl and mouse recordings, the activity of the ganglion cells was amplified, band-pass filtered (300 Hz to 5 kHz), and recorded digitally at 25 kHz (60-electrode arrays) or 10 kHz (252-electrode arrays) using the MC-Rack software (Version 4.6.2, MultiChannel Systems). The spikes were extracted from the recorded voltage traces with a custom-made spike-sorting program based on a Gaussian mixture model and an expectation-maximization algorithm [60].

The recordings from marmoset retina were performed as described previously [61]. Briefly, eyes were quickly enucleated and dissected to allow direct supply to the retina with oxygenated (95% O2 and 5% CO2) Ames' medium (Sigma-Aldrich, Munich, Germany), supplemented with 6 mM D-glucose, and buffered with 22 mM NaHCO3 to maintain a pH of 7.4. Retinas were then dark-adapted for 1–2 hours. For each recording, a small peripheral retina piece, with pigment epithelium removed, was placed onto a 4096-channel CMOS-based microelectrode array (21 µm electrode size and 42 µm pitch; 3Brain AG). The retina piece was mounted with the ganglion cell-side down, covered with a piece of translucent polyester filter membrane (Pieper Filter #PE0104700; 0.1 µm pores), and weighted down with a metal ring. During the recording, the retina was perfused with the oxygenated Ames' medium (4–5 ml/min), and the temperature of the recording chamber was kept constant at around 33°C in the same way as for the mouse recordings. Spike extraction and sorting were performed by HerdingSpikes2 (github.com/mhhennig/HS2). In short, spiking events were detected independently at each electrode using a fixed threshold, then simultaneous events at neighboring electrodes were triangulated to identify the location and shape of the spike. Finally, events were clustered into units, using a mean-shift algorithm, based on the location and first two principal components of the spike waveform [62,63]. Repeated units were detected and removed, based on the similarity between their responses to white-noise stimuli (see next section for stimulus details). To that end, the firing rate of each cell was binned according to the stimulus frames, then the correlation coefficient between the binned firing rates (with no time lag) was calculated for all pairs of cells within a simultaneously recorded population. Next, a graph of the simultaneously recorded cells was constructed with one node for each cell and one edge between two nodes if the response correlation between the two cells was > 0.3. Repeated cells were detected by finding cliques within this graph. Within each group of repeated cells, all except one were removed.

All retina preparations were performed under infrared illumination with a stereomicroscope equipped with night-vision goggles.

## Visual stimulation

Visual stimuli for all recordings were generated and controlled by custom-made software written in C++ and using the OpenGL library. For both axolotl and marmoset recordings, the stimuli were presented on a monochromatic gamma-corrected white OLED monitor (eMagin, 800×600 pixels, 60 Hz refresh rate, 75 Hz in a few axolotl experiments). For the mouse recordings, the stimuli were displayed via modified DLP Lightcrafter projectors (evaluation module, 864×480 pixels, 60 Hz refresh rate, Texas Instruments Company, Dallas, USA), with the blue LED replaced by a UV LED (emission peak at 365 nm), as described previously [59]. Sizes of the projections of monitor pixels on the retina were 2.5 μm×2.5 μm for axolotl, 7.5 μm×7.5 μm for marmoset, and 8 μm×8 μm for mouse. All stimuli were presented with a mean light intensity in the mesopic to low-photopic range (2.5 mW/m$^2$ for axolotl, 0.75 to 2.8 mW/m$^2$ for marmoset, and 5.06 to 5.2 mW/m$^2$ for mouse).

The white-noise stimulus was obtained by drawing new screen intensities randomly from a normal distribution at either 30 Hz (axolotl; except for 25 Hz and 37.5 Hz in one recording each) or 60 Hz (mouse and marmoset). The normal distribution of screen intensities was centered at half of the maximum intensity of the display and had a standard deviation of 30% of the mean light intensity. In the recordings from axolotl and mouse retinas, a fixed white-noise sequence of 150 or 300 intensity values ("frozen noise") was repeatedly inserted after every 750, 900, or 1500 intensity values (with variations coming from different recording sessions) of non-repeating white noise. For model evaluation, the data was subdivided into training and held-out test data. When frozen noise data was available (axolotl and mouse data), it was held out as the test data. Otherwise (marmoset data), 6.7 s of data were taken from every 33.3 s as held-out pseudo-trials for testing. The total amount of data varied between experiments, but was typically around 7 min for training and 2 min for testing (average over all analyzed cells: 427±187 s (mean±SD) for training, 113±46 s for testing).

The contrast-step stimulus, the contrast sweep, and the frequency sweep were presented consecutively, in a combo called the chirp stimulus [20]. In each trial of the chirp stimulus, following 120 frames (i.e., 2 s) of mean (background) light intensity (0% contrast), ON and OFF stimuli (+100% and -100% contrast, respectively) were presented for 60 frames each. After another 60 frames at background intensity, the sinusoidal frequency sweep was presented, starting at 0 Hz and linearly ramping up to 15 Hz over a duration of 480 frames, fluctuating around the mean stimulus intensity with peak light levels at -100% and +100% contrast. After another 60 background-intensity frames, the sinusoidal contrast sweep was presented, starting at 0% peak contrast and linearly ramping to 100% at a frequency of 4 Hz over a duration of 480 frames. At the end, there were another 60 background-intensity frames. The chirp stimulus was repeated for 15 trials. For analysis of ganglion cell responses, spike trains were binned according to the stimulus frames at 60 Hz and averaged over trials to obtain time-dependent firing rates. The contrast-step part of the chirp stimulus was used for classifying cells into transient and sustained cells. To do so, the ratio of sustained response rate (computed as the average firing rate during the last 700 ms of the preferred contrast presentation) to initial response peak (obtained as the peak firing rate within the first about 200 ms of the same contrast presentation) was computed, and cells with a ratio below 0.2 were considered as transient, otherwise as sustained. The frequency and contrast sweeps were used for evaluating the generalization of the models to a range of stimulus frequencies and contrasts.

## Model construction and training

All model analyses were performed with time bins discretized at the update rate of the stimulus and responses given by the number of spikes in each time bin. Each model contains one branch (for the feedback and the LN model) or two convergent branches (for the subtractive and the divisive model) that represent upstream computation. After combining the two branches additively for the subtractive model and multiplicatively for the divisive model, the upstream computation in each model is followed by an output rectifier and a spike generator. Each upstream branch consists of a stimulus filter and a subsequent nonlinear function. The feedback model's feedback loop consists of a filter that is applied to the recent history of spiking activity with no nonlinear function (Fig 1A). To maximize the consistency in implementation

and training, we combined all four models in one master structure (S1 Fig). The input to this master structure was the one-dimensional time series of numbers used to construct the contrast values of the stimulus (drawn from a normal distribution of zero mean and unit standard deviation), with zero corresponding to mean light intensity. Each filter was parametrized by sampling at the stimulus update rate, using 30 samples for axolotl, 25 for mouse, and 15 for marmoset. Each upstream nonlinear function was parametrized with a set of tent-shaped basis functions as done previously [11], here using 15 basis functions, spanning the input range [-3, 3]. The output rectifier was described as a "softplus" function, $f(x) = m\ln(1 + e^{ax+b}) + c$, with $a$, $b$, $c$, and $m$ representing free parameters.

In the unified structure, only one model component, i.e., a filter, an upstream nonlinearity, or the rectifier function, was trained at a time. To this end, we write the output of this structure as the expansion in the basis functions of the branch that is under training, while describing the rest of the model by a set of fixed contributions, e.g.,:

$$r(t) = f\left(\left[\pm \sum_i \left(w_i\, g_i \left(\sum_\tau (k_\tau X(t-\tau))\right)\right) \pm P(t)\right] \times M(t) + A(t)\right)$$

in which $k_\tau$ are the elements of the filter under training and $w_i$ the weights for the basis functions of the corresponding nonlinear function, $f$ is the output rectifier, and $g_i$ are the pre-specified basis functions of the nonlinearities. The sign in front of the signal of a filter branch was fixed to be negative for the suppressive branch of the subtractive model and positive for all other model branches.

For training each branch of each model, variables $X$, $P$, $M$, and $A$ were defined from Table 1. This table shows, for example, that when the excitatory branch of the subtractive model was being updated, it was represented as the weighted sum in the formula above, while the suppressive branch was represented by variable $P$ and vice versa. Similarly, for the divisive model, when the excitatory branch was being updated, the suppressive branch was represented by M and vice versa. Note that M is multiplicative in the formula, but acts as divisive suppression, because the nonlinearity of the suppressive branch is constrained to be bump-shaped with a maximum of unity for a filter output of zero.

The branch being trained was updated using a block-coordinate descent algorithm, in which the filters $k_\tau$ were updated to minimize the negative log-likelihood function while the weights $w_i$ were constant and vice versa. The components of the model were trained in this order: upstream excitatory, upstream suppressive (for subtractive and divisive models), the feedback loop (for the feedback model), and the output rectifier.

The objective function was the negative of the log-likelihood, $-LL$, and was calculated from the natural logarithm of the total probability that the number of spikes in each time bin $t$ would be drawn from a Poisson distribution with an expectation value of $r(t)$ for each bin $t$. In each training iteration, we used the function fmincon in Matlab to take 10 steps per model component toward minimizing the objective function. We trained each model for either 100 iterations or until the change in the objective function was smaller than 0.01%. In fmincon, we used the 'trust region reflective' algorithm. The

**Table 1. Assignment of the variables to derive each model from the unified structure.**

| Model | Components | X | P | M | A |
|---|---|---|---|---|---|
| **Subtractive** | exc (Excitatory) | stim | sup output | 1 | 0 |
| | sup (Suppressive) | stim | exc output | 1 | 0 |
| **Divisive** | exc | stim | 0 | sup output | 0 |
| | sup | stim | 0 | exc output | 0 |
| **Feedback** | exc | stim | 0 | 1 | feedback output |
| | sup | spikes | 0 | 1 | exc output |
| **LN** | exc | stim | 0 | 1 | 0 |

constraints on the $w_i$ were applied using the sequential quadratic programming algorithm. The derivative of the objective function was provided to fmincon by calculating the derivatives of $-LL$ with respect to the $w_i$ as well as to the $k_\tau$ and to the free parameters of $f(\cdot)$.

The constraints for the optimization of each parameter are specified in Table 2. In particular, filters were required to be normalized (Euclidean norm equal to unity) and tail off towards zero. The nonlinearity weights $w_i$ were constrained in such a way that the upstream nonlinearities of the excitatory branches and of the suppressive branch in the subtractive model were monotonically rising and that the upstream nonlinearity of the suppressive branch in the divisive model was bump-shaped, that is, monotonically rising for negative input and monotonically decaying for positive input. These constraints had also been applied in the original formulations of the models [1,11] and support the interpretation as excitatory and suppressive filter branches, respectively. The monotonic shape in the suppressive branch of the subtractive model, for example, provides for increasing suppression with increasing activation of the corresponding filter, as the sign of the contribution of this branch is fixed to be negative. Equivalently, the excitatory branches provide stronger excitation for stronger filter activation because of their monotonic nonlinearity and the positive sign assigned with the branches. For the divisive model, on the other hand, the bump-like shape guarantees that any filter activation provides more (or at least as much) suppression as compared to smaller activation of the same sign or no filter activation. Note that requiring a monotonically decaying shape of the nonlinearity or simply constraining the maximum of the nonlinearity by unity is not sufficient for keeping the suppressive nature of this branch. This is because the sign of the filter is ambivalent (allowing to turn a decreasing into an increasing nonlinearity by flipping the sign of each filter element) and because the absolute values of the nonlinearities could be scaled up or down if the inverse scaling is performed on the nonlinearity of the excitatory branch. (This potential model redundancy is in fact addressed by constraining the maximum of the bump-shaped nonlinearity to unity.) Thus, the bump-shaped nonlinearity is an essential feature of the model with divisive suppression.

To reduce the risk of finding non-optimal solutions corresponding to local minima of the objective function, the training was repeated 5 times with different initializations that were guided by simple spike-triggered analyses. We then selected the solution for which the value of the objective function (negative log-likelihood) for the training data was the lowest. The initial values were selected from the spike-triggered average, the most relevant stimulus features extracted from a spike-triggered-covariance analysis [64,65], pre-defined deterministic functions and Gaussian noise, as specified in Table 3. (When different initial values were used for each model, the entries in the table follow the format of sub/ div/ fb/ LN.) Here, STA was the spike-triggered average, $STC_1$ and $STC_N$ were the first and the last principal components of the spike-triggered covariance (with the STA projected out from the stimulus segments before the covariance analysis), N(0,1) was Gaussian white noise with mean of zero and standard deviation of unity, $W_{mono}$ was the softplus function, as defined above, with $a = 10$, $b,c = 0$, and $m = 0.1$, $W_{bi}$ was a bell-shaped function (Gaussian spanning approximately 2 standard deviations over the range [-3, 3], normalized to a magnitude range of [0, 1]). To add variability in the initialization, Gaussian

**Table 2. The optimization constraint for the set of variables of each model component. *The tail of the filter was the average of the values of the last five time bins.**

| Model | Exc filter | Exc function | Sup filter | Sup function |
|---|---|---|---|---|
| Subtractive | Norm = 1<br>Tail* = 0 | Monotonically increasing<br>Lower bound = $10^{-16}$ | Norm = 1<br>Tail = 0 | Monotonically increasing<br>Lower bound = $10^{-16}$ |
| Divisive | Norm = 1<br>Tail = 0 | Monotonically increasing<br>Lower bound = $10^{-16}$ | Norm = 1<br>Tail = 0 | Monotonically increasing for input < 0<br>& decreasing for input > 0<br>Scaled to [$10^{-16}$, 1] |
| Feedback | Norm = 1<br>Tail = 0 | Monotonically increasing<br>Lower bound = $10^{-16}$ | Tail = 0 | – |
| LN | Norm = 1<br>Tail = 0 | Monotonically increasing<br>Lower bound = $10^{-16}$ | – | – |

**Table 3. Initial values for 5 training runs.**

| Run number | Exc filter | Exc function | Sup filter | Sup function |
|---|---|---|---|---|
| 1 | STA | $W_{mono}$ | $STC_1$/ $STC_1$/ 0/ - | $W_{mono}$/ $W_{bi}$/ -/ - |
| 2 | STA | $W_{mono}$/ $W_{bi}$/ $W_{mono}$/ $W_{bi}$ | $STC_N$/ $STC_N$/ 0/ - | $W_{mono}$/ $W_{bi}$/ -/ - |
| 3 | $STC_1$ | $W_{mono}$ | $STC_N$/ $STC_N$/ 0/ - | $W_{mono}$/ $W_{bi}$/ -/ - |
| 4 | $STC_N$ | $W_{mono}$/ $W_{bi}$/ $W_{mono}$/ $W_{bi}$ | $STC_1$/ $STC_1$/ 0/ - | $W_{mono}$/ $W_{bi}$/ -/ - |
| 5 | $N(0,1)$ | $W_{mono}$ | $N(0,1)$ | $W_{mono}$/ $W_{bi}$/ -/ - |

noise with a mean of zero and standard deviation of 0.1 was added to the original output values of $W_{mono}$ and $W_{bi}$ at each time. For all models and training runs, the output rectifier was initialized as $10 \cdot \ln(1 + e^{0.1x})$.

## Evaluation of the fitted models

The best solution of each model for each cell was finally evaluated on the held-out test data. To do so, the model performance was assessed by calculating the average information per spike provided by the model about the cell's spiking response [2,16]. The information per spike was calculated as

$$\frac{LL\ (\hat{r}|r) - LL\ (\bar{r}|r)}{\bar{r}\ \ln(2)}$$

Here, $r$ denotes the spike train data (spikes per bin), $\bar{r}$ the corresponding average, $\hat{r}$ the prediction by the evaluated model, and $LL$ the likelihood of the fitted model and for a constant Poisson rate, respectively. The denominator ensures normalization to obtain the information in bits/spike. Note that this measure of model performance does not require repeated trials. For the feedback model, the performance was averaged across 100 evaluations, each with spike counts obtained stochastically according to a Poisson distribution with expectation value given by the model's output.

Poisson explained variance was used as a secondary measure of the model performance when firing rates over repeated trials of frozen noise were available as held-out test data. This was computed as

$$\text{Poisson explained var} = 1 - \text{dev}_{Poisson}\ (r,\ \hat{r}) / \text{dev}_{Poisson}\ (r,\ \bar{r})$$

in which $\text{dev}_{Poisson}$ was the Poisson deviance, calculated as

$$\text{dev}_{Poisson}\ (r,\ \hat{r}) = 2\sum \left( \ln\left( \left(\frac{r}{\hat{r}}\right)^r \right) - (r - \hat{r}) \right)$$

where the sum runs over time bins. Here, $r$ is the trial-averaged spike count per time bin, $\hat{r}$ the corresponding model prediction, and $\bar{r}$ the average firing rate (per time bin) over the frozen-noise section.

The performance of the models for the frequency and contrast sweep stimuli were computed using the explained variance ($R^2$) according to

$$R^2 = 1 - \frac{\sum (r - \hat{r})^2}{\sum (r - \bar{r})^2}$$

Here, again, the sums run over all time bins of the analyzed time window as specified in the main text, $r$ is the trial-averaged firing rate per time bin, $\hat{r}$ the corresponding model prediction, and $\bar{r}$ the average firing rate over the analyzed time window.

## Selection of units

We selected units for further analysis based on the following criteria: 1. The average firing rate of the unit under the white-noise stimulus was required to be larger than 2 Hz for axolotl and larger than 5 Hz for mouse and marmoset. 2. Responses to the frozen-noise presentation for units from axolotl and mouse were required to be sufficiently reliable across trials. To this end, response reliability was assessed by dividing the responses of a cell into two subsets of equal sizes, computing a peri-stimulus time histogram (PSTH) separately for each subset, and calculating the variance of the first PSTH explained by the second. The reliability was obtained as the average explained variance across 20 random subset divisions and was required to be larger than 0.5. 3. The difference between the average firing rate in the early (first 30% of the stimulus duration) and late (last 30% of the stimulus duration) part of the recording under the white-noise stimulus was required to be less than 50% of the overall average firing rate. 4. The unit was required to not correspond to an ON-OFF cell, i.e., it should not respond to both light increments and light decrements. ON-OFF cells were detected by a U-shape nonlinear function of the LN model: The function was divided into the left (input $< 0$) and the right (input $> 0$) part, then a line was fitted to the left (slope = sL) and another line was fitted to the right part (slope = sR). A cell was classified as ON-OFF, if $sL/(|sL|+|sR|) < -0.2$. 5. Performance on the test data was required to be at least 60% of the performance on the training data for all four models to reduce the effects of overfitting.

## Statistical analysis

Two-sided Wilcoxon signed-rank test was used, except where another test was indicated. When multiple groups of data were compared, false discovery rate (FDR) correction for multiple comparisons [19] was used to correct the $p$-values.

## Supporting information

**S1 Fig. The unified model structure that incorporates all components of the models in Fig 1A.** The filter components are shown with thin frames and the nonlinearities with thick frames. Exc upstream branch represents the excitation, $Sup_1$ represents the subtractive suppression, and $Sup_2$ represents the divisive suppression. The feedback (fb) was added to the downstream part of the model.
(TIF)

**S2 Fig. Comparing the performance of our trained LN model to the performance of an LN model that uses the STA as the filter (LN0) or the first principal component of the STC (LN1).** For both LN0 and LN1, the nonlinear function was computed as the histogram of filter outputs versus spike counts. For the comparison here, either LN0 or LN1 was chosen, depending on which model had better performance.
(TIF)

**S3 Fig. Same as Fig 1B and 1C for the axolotl and mouse data, but showing Poisson explained variance (exp var), as measured on the frozen-noise sections of the data. A)** Comparison of the Poisson explained variance of each model to the explained variance of the LN model. Each data point represents one cell. **B)** *Left*: The excess explained variance (computed as explained variance of suppressive model minus explained variance of the LN model) of the best performing suppressive model, determined for each cell. *Right*: The percentage of cells for which the corresponding suppressive model outperformed all other models in terms of explained variance.
(TIF)

**S4 Fig. Principal component analysis of temporal filter shapes to separate cells recorded from axolotl retina into functional classes. A)** The eigenvalues associated with the 30 principal components of the set of filters of the LN model. *Inset:* The first and the second principal components, which were used for classification. **B)** *Center:* All analyzed axolotl cells in the space of $s_1$, the score for the first principal component ($comp_1$), and $s_2$, the score for the second principal

component ($comp_2$). To determine the classes, the scores for each cell were compared to thresholds of ±0.02: for slow ON cells (n = 48) $s_1 > 0.02$ and $s_2 > 0.02$, for fast ON cells (n = 13) $s_1 > 0.02$ and $s_2 < -0.02$, for slow OFF cells (n = 195) $s_1 < -0.02$ and $s_2 < -0.02$ and for the fast OFF cells (n = 208) $s_1 < -0.02$ and $s_2 > 0.02$. 143 cells remained unclassified, as they did not pass any of the threshold combinations. *Insets surrounding the scatter plot:* LN-model filters (thin lines) for each of the identified groups, together with their average (thick line).
(TIF)

**S5 Fig. The responses (black lines) in spikes per second (sp/s) of the sample cells from Fig 5A-D and the model predictions (colored lines) for the entire ranges of the contrast sweep and the frequency sweep.** The stimulus traces (gray lines on top) are displayed with temporal sampling corresponding to the monitor update. **A)** The cell from Fig 5A and 5B for the contrast sweep. **B)** The cell from Fig 5C and 5D for the frequency sweep.
(TIF)

## Author contributions

**Conceptualization:** Neda Shahidi, Tim Gollisch.

**Data curation:** Neda Shahidi, Fernando Rozenblit, Mohammad H. Khani, Helene M. Schreyer.

**Formal analysis:** Neda Shahidi.

**Funding acquisition:** Tim Gollisch.

**Investigation:** Neda Shahidi, Fernando Rozenblit, Mohammad H. Khani, Helene M. Schreyer, Matthias Mietsch, Dario A. Protti, Tim Gollisch.

**Methodology:** Neda Shahidi, Fernando Rozenblit, Mohammad H. Khani, Helene M. Schreyer, Matthias Mietsch, Dario A. Protti.

**Project administration:** Tim Gollisch.

**Resources:** Matthias Mietsch, Tim Gollisch.

**Software:** Neda Shahidi, Fernando Rozenblit, Mohammad H. Khani, Tim Gollisch.

**Supervision:** Tim Gollisch.

**Visualization:** Neda Shahidi, Tim Gollisch.

**Writing – original draft:** Neda Shahidi, Tim Gollisch.

**Writing – review & editing:** Neda Shahidi, Fernando Rozenblit, Mohammad H. Khani, Helene M. Schreyer, Matthias Mietsch, Dario A. Protti, Tim Gollisch.

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
