## [Decision Letter · Decision Letter 0]

12 Dec 2024

PCOMPBIOL-D-24-01569

Filter-based models of suppression in retinal ganglion cells: comparison and generalization across species and stimuli

PLOS Computational Biology

Dear Dr. Gollisch,

Thank you for submitting your manuscript to PLOS Computational Biology. After careful consideration, we feel that it has merit but does not fully meet PLOS Computational Biology's publication criteria as it currently stands. Therefore, we invite you to submit a revised version of the manuscript that addresses the points raised during the review process.

Please submit your revised manuscript within 30 days Feb 11 2025 11:59PM. If you will need more time than this to complete your revisions, please reply to this message or contact the journal office at ploscompbiol@plos.org. Please include the following items when submitting your revised manuscript:

We look forward to receiving your revised manuscript.

Kind regards,

Matthias Helge Hennig, Ph.D.

Academic Editor

PLOS Computational Biology

Daniele Marinazzo

Section Editor

PLOS Computational Biology

**Journal Requirements:**

At this stage, the following Authors/Authors require contributions: Neda Shahidi, Fernando Rozenblit, Mohammad H. Khani, Helene M. Schreyer, Matthias Mietsch, Dario A. Protti, and Tim Gollisch. Please ensure that the full contributions of each author are acknowledged in the "Add/Edit/Remove Authors" section of our submission form.

2) We ask that a manuscript source file is provided at Revision. Please upload your manuscript file as a .doc, .docx, .rtf or .tex. If you are providing a .tex file, please upload it under the item type LaTeX Source File and leave your .pdf version as the item type Manuscript.

5) We notice that your supplementary Figures are included in the manuscript file. Please remove them and upload them with the file type 'Supporting Information'. Please ensure that each Supporting Information file has a legend listed in the manuscript after the references list.

6) Some material included in your submission may be copyrighted. According to PLOS copyright policy, authors who use figures or other material (e.g., graphics, clipart, maps) from another author or copyright holder must demonstrate or obtain permission to publish this material under the Creative Commons Attribution 4.0 International (CC BY 4.0) License used by PLOS journals. Please closely review the details of PLOSu2019s copyright requirements here: PLOS Licenses and Copyright. If you need to request permissions from a copyright holder, you may use PLOS's Copyright Content Permission form.

Potential Copyright Issues:

i) Figures 1A, and S1. Please confirm whether you drew the images / clip-art within the figure panels by hand. If you did not draw the images, please provide (a) a link to the source of the images or icons and their license / terms of use; or (b) written permission from the copyright holder to publish the images or icons under our CC BY 4.0 license. Alternatively, you may replace the images with open source alternatives. See these open source resources you may use to replace images / clip-art:

7) Thank you for uploading your study's underlying data set. We notice that there is a CC BY-SA 4.0 license on your data. We would encourage you to consider using a license that is no more restrictive than CC BY, in line with PLOS’ recommendation on licensing (http://journals.plos.org/plosone/s/licenses-and-copyright). For a list of recommended repositories and additional information on PLOS standards for data deposition, please see https://journals.plos.org/ploscompbiol/s/recommended-repositories

**Reviewers' comments:**

Reviewer's Responses to Questions

Reviewer #1: In this study the authors compare the performance of several models of suppression at explaining quantitatively how ganglion cells respond to random fluctuations of contrast, for a stimulus that is spatially uniform. They find that suppressive and divisive models are better than a spike history filter model. This is a solid study with interesting results.

My concerns:

1) My understanding is that the feedback model is just implementing the kind of spike history filter usually found in GLM models (Pillow et al). There are other models that perform at this, and have been used before. For example, the gain control model used in Berry et al, 1999, but also in Chen et al, 2013,2014 ; Leonardo and Meister, etc. This gain control model has a divisive component. It seems necessary to compare the performances to this type of model too.

2) The main result is that the divisive and the subtractive models are not doing exactly the same thing. Their performance depend on stimulus frequency, a result I find very interesting. It would be nice to have more intuition about why this is the case.

3) A caveat of this study, acknowledged by the authors, is that everything they do is restricted to spatially uniform stimuli. The lack of substantial difference between the divisive and subtractive model could come from this. I don’t expect the authors to address the issue of spatially detailed stimuli, but a bit more discussion on this would be helpful, especially given the other works in the field.

4) Related to this, the discussion contains very little discussion about other works in the field (eg ref 1 of the paper). This is an issue to better identify the novelty of this work.

5) Some parts of the paper are a bit unclear. In particular, the paragraph that includes the sentence: In the divisive model, on the other hand, the shape of the nonlinearity in the suppressive branch led to a unilateral, rectified suppression with essentially no sensitivity to filter activations near to or smaller than zero

is hard to understand for me and deserves a proper explanation. The first panel of the figure 3 might benefit from being expanded. The non-linearity is learned but this is not obvious from the text. Also, the NL learning seems to hit the bounds imposed. What happens if they are pushed back ?

Reviewer #2: This is a thorough and well-executed exploration of different classes of combined excitation and suppression in neurons. The work focuses on retinal ganglion cells and taps three species (axolotl, mouse, marmoset) to add a comparative angle to the study. Models are well-characterized and compared for two main classes of input stimuli - full-field flicker, chirps, and contrast sweeps. Results show that divisive suppression is the most versatile motif, with good reproduction of responses across the widest range of stimuli. Subtractive suppression was only the clear winner in slow axolotl cells. A large number of recordings were made, the methods are sound and well-explained, and the text flows well and is well-referenced.

Major comment:

The only way this paper could be found lacking is in a clearer or more thoroughly explored functional consequence to the findings. The Discussion certainly hints at some ideas surrounding the differences in these types of suppression, but more could be made about:

1) the differences between axolotl and mouse/primate: is this a result of "more processing" happening in the amphibian retina compared to mammalian species? Does this connect to behavioral repertoire or environmental statistics in some potentially interesting way?

2) how divisive suppression might be better suited to dynamic environments. Some of the Discussion touches on adaptive mechanisms, but a bit more could be said here.

This points could better expand the reach of the work beyond the retinal community, though the work stands as a useful and solid set of insights for cortical neurophysiologists. A more explicit connection to non-retinal processing in the Discussion could be useful.

Minor comment:

On page 10, is "motives" meant to be "motifs"?

**Have the authors made all data and (if applicable) computational code underlying the findings in their manuscript fully available?**

Reviewer #1: Yes

Reviewer #2: Yes

PLOS authors have the option to publish the peer review history of their article (what does this mean? ). If published, this will include your full peer review and any attached files.

**Do you want your identity to be public for this peer review?** For information about this choice, including consent withdrawal, please see our Privacy Policy .

Reviewer #1: No

Reviewer #2: No

**Figure resubmission:**
---

## [Decision Letter · Decision Letter 1]

7 Apr 2025

Dear Gollisch,

We are pleased to inform you that your manuscript 'Filter-based models of suppression in retinal ganglion cells: comparison and generalization across species and stimuli' has been provisionally accepted for publication in PLOS Computational Biology.

Best regards,

Matthias Helge Hennig, Ph.D.

Academic Editor

PLOS Computational Biology

Daniele Marinazzo

Section Editor

PLOS Computational Biology

Reviewer's Responses to Questions

**Comments to the Authors:**

Reviewer #1: The authors have addressed all my questions.

Reviewer #2: I am very pleased with the extensive revisions made to the text. The authors have addressed all of my concerns.

**Have the authors made all data and (if applicable) computational code underlying the findings in their manuscript fully available?**

Reviewer #1: None

Reviewer #2: Yes

PLOS authors have the option to publish the peer review history of their article (what does this mean? ). If published, this will include your full peer review and any attached files.

**Do you want your identity to be public for this peer review?** For information about this choice, including consent withdrawal, please see our Privacy Policy .

Reviewer #1: No

Reviewer #2: No

---

## [Editor Report · Acceptance letter]

PCOMPBIOL-D-24-01569R1

Filter-based models of suppression in retinal ganglion cells: comparison and generalization across species and stimuli

Dear Dr Gollisch,

I am pleased to inform you that your manuscript has been formally accepted for publication in PLOS Computational Biology. Your manuscript is now with our production department and you will be notified of the publication date in due course.

With kind regards,

Anita Estes
